# Effects of Different Titanium Surface Treatments on Adhesion, Proliferation and Differentiation of Bone Cells: An In Vitro Study

**DOI:** 10.3390/jfb13030143

**Published:** 2022-09-05

**Authors:** Milan Stoilov, Lea Stoilov, Norbert Enkling, Helmut Stark, Jochen Winter, Michael Marder, Dominik Kraus

**Affiliations:** 1Department of Prosthodontics, Preclinical Education and Dental Materials Science, Medical Faculty, University of Bonn, 53111 Bonn, Germany; 2Department of Reconstructive Dentistry and Gerodontology, School of Dental Medicine, University of Bern, 3012 Bern, Switzerland; 3Department of Periodontology, Operative and Preventive Dentistry, Medical Faculty, University of Bonn, 53111 Bonn, Germany

**Keywords:** dental implants, titanium surface, osteoblasts, cell adhesion, cell proliferation, cell differentiation

## Abstract

The objective of this study was to evaluate the impacts of different sandblasting procedures in acid etching of Ti6Al4V surfaces on osteoblast cell behavior, regarding various physicochemical and topographical parameters. Furthermore, differences in osteoblast cell behavior between cpTi and Ti6Al4V SA surfaces were evaluated. Sandblasting and subsequent acid etching of cpTi and Ti6Al4V discs was performed with Al_2_O_3_ grains of different sizes and with varying blasting pressures. The micro- and nano-roughness of the experimental SA surfaces were analyzed via confocal, atomic force and scanning electron microscopy. Surface free energy and friction coefficients were determined. hFOB 1.19 cells were seeded to evaluate adhesion, proliferation and osteoblastic differentiation for up to 12 d via crystal violet assays, MTT assays, ALP activity assays and Alizarin Red staining assays. Differences in blasting procedures had significant impacts on surface macro- and micro-topography. The crystal violet assay revealed a significant inverse relationship between blasting grain size and hFOB cell growth after 7 days. This trend was also visible in the Alizarin Red assays staining after 12 d: there was significantly higher biomineralization visible in the group that was sandblasted with smaller grains (F180) when compared to standard-grain-size groups (F70). SA samples treated with reduced blasting pressure exhibited lower hFOB adhesion and growth capabilities at initial (2 h) and later time points for up to 7 days, when compared to the standard SA surface, even though micro-roughness and other relevant surface parameters were similar. Overall, etched-only surfaces consistently exhibited equivalent or higher adhesion, proliferation and differentiation capabilities when compared to all other sandblasted and etched surfaces. No differences were found between cpTi and Ti6Al4V SA surfaces. Subtle modifications in the blasting protocol for Ti6Al4V SA surfaces significantly affect the proliferative and differentiation behavior of human osteoblasts. Surface roughness parameters are not sufficient to predict osteoblast behavior on etched Ti6Al4V surfaces.

## 1. Introduction

Successful osseointegration after implantation is regarded as one of the most decisive factors for long-term survival of oral implants [1,2] and implants in general. It is determined by a variety of factors, including implant material biocompatibility and surface characteristics [3,4]. In this context, various surface modifications for titanium and titanium alloy implants are currently employed that aim to enhance the biological reaction between vital tissue and the implant surface, and to maximize new bone formation in the bone-to-implant interface after implantation.

Commercially pure titanium (cpTi) and its major medical alloy, Ti6Al4V, are widely used for biomedical implants because of their biocompatibility, mechanical properties and neutral interference during modern imaging techniques (CT, MRI). Both cpTi and Ti6Al4V develop a highly stable surface oxide layer when exposed to air or to aqueous media, which is responsible for the bone-bonding characteristics of titanium implants [5]. Although cpTi is the material of choice in dental implantology [6], its use is limited in areas subjected to high wear and tensile and fatiguing loads [7,8,9]. As titanium is a relatively soft material [10], fatigue may occur, particularly when it is used in small-diameter implants which must fulfill high requirements for mechanical stability to avoid overload and implant fracturing [9,11]. Ti alloys such as Ti6Al4V are thus widely employed as alternatives, owing to their excellent mechanical properties [10,12]. Nevertheless, Ti6Al4V is suspected to also exhibit negative effects on cell viability by releasing Al^3+^ and V^2+^ ions [13,14,15] into surrounding tissues, having possible adverse effects on implant biocompatibility [16].

In general, osteoblast attachment to smooth titanium or titanium alloy surfaces is restricted, and faster osseointegration has been observed around modified surfaces, when compared to smooth, machined or polished surfaces [17]. One of the most widely applied surface modifications for titanium and titanium alloy implants is a combination of large-grit sandblasting and acid-etching, termed SA, where the surface is first blasted with coarse abrasive particles and is subsequently subjected to dual acid-etching with various strong acids. This treatment creates an isotropic topography [18] that is characterized by irregularities on the macro-scale and intercommunicated cavities at the micron and sub-micron scale. Improved osseointegrative characteristics of these surfaces are thought to result from enhanced mechanical interlocking with the surrounding bone and the increased surface area, surface energy, protein adsorption and cell adhesion [19,20,21] in the initial wound healing process. Moreover, micro-roughened titanium surfaces were found to induce differences in proliferation, differentiation and the secretory profile of osteogenic cells [22,23] when compared to machined implants.

Historically, the effect of those roughened surface modifications has been directly associated with the change in topography [1,24]. The most common parameters for determining surface roughness are the two-dimensional (Sa) and one dimensional (Ra) arithmetical mean height. Sa is the extension of Ra (arithmetical mean height of a line) to a surface. It expresses, as an absolute value, the difference in height of each point compared to the arithmetical mean of the surface. Micro roughness (Sa) was historically subdivided into different classes: smooth (<0.5 µm), minimally rough (0.5–1.0 µm), moderately rough (1.0–2.0 µm) and highly rough (>2.0 µm) surfaces. Notably, moderately rough surfaces with an arithmetic mean roughness of 1.2–2 µm exhibit excellent osseointegration in in-vivo studies [18,25] and have been commonly used for dental implants for more than a decade [26,27].

However, aside from the parameters of mean surface roughness (Ra/Sa), scientists have struggled to link other topographical parameters at the micron scale to the biological response [28]. To add to the complexity, modification techniques such as SA that cause topographical changes at the micrometer scale may also affect the surface in other dimensions or alter the physicochemical properties. Moreover, topographical and physicochemical properties of the implant surfaces are often inherently inter-related [29], rendering it difficult to attribute a direct causative effect to isolated topographical parameters. In consequence, the focus of titanium surface modifications at the micron scale has recently expanded to include significant parameters—e.g., hydrophilicity and nano-roughness.

The objective of this study was to evaluate the influence of the sandblasting procedure in standard SA Ti6Al4V surfaces on osteoblast cell behavior, in consideration of various physicochemical and topographical parameters, such as wettability, hydrophilicity, friction coefficient and micro- and nano-roughness. Blasting pressure and Al_2_O_3_ grain size were varied in the procedure, and the effects on osteoblast adhesion, proliferation and differentiation were evaluated.

## 2. Materials and Methods

### 2.1. Surface Modifications

A total of 480 test samples were provided by Dentaurum GmbH&Co.KG (Ispringen, Germany) from the following materials: pure titanium grade 4 (cpTi, n = 80) and alloyed titanium grade 5 (Ti6Al4V; n = 400). The standardized titanium discs were manufactured in a highly standardized and industrial process and had a thickness of 1 mm and a diameter of 10 mm. On the basis of a sandblasted and acid-etched surface (SA surface), the sandblasting protocol was varied by modification of blasting pressure and grain size (Table 1). The standardized pressure of sandblasting and the acid-etching protocol represent non-published, internal company procedures of Dentaurum GmbH&Co.KG.

### 2.2. Analysis of Surface Topography

The titanium samples (groups 2.1–2.5; each n = 2) were examined by the Steinbeis Transfer Center (STZ-Tribology; Karlsruhe, Germany) regarding surface topographies, prior to in vitro cell culture experiments. The following analytical methods were performed: confocal laser scanning microscopy (CLSM), atomic force microscopy (AFM), scanning electron microscopy (SEM) and measurements of the surface free energy and the friction coefficient. Using CLSM (LEXT OLS 4100, Olympus Europa SE and Co. KG, Hamburg, Germany) and AFM (LensAFM V2, Nanosurf AG, Liestal, Switzerland), the titanium surfaces were measured at the micro- (definition area 2 × 250 µm × 200 µm; Gaussian filter 50 × 50 µm) and nanometer scales (definition area 2 × 5 µm × 5 µm; Gaussian filter 0.25 × 0.25 µm). Furthermore, samples were scanned in profile (1.4 mm) for obtaining one-dimensional surface parameters (R) and in definition areas for two-dimensional surface parameters (S) with Gaussian filters. Thus, surfaces were examined both one- and two-dimensionally (R and S values, respectively). Additional surface height parameters, Sz (maximal height deviation), Sq (root mean square height) and developed interfacial area ratio (Sdr), and the spatial parameter Sds (density of peaks), were determined by CLSM (according to ISO 25178). Topography was analyzed via SEM (Evo M10, Carl Zeiss AG, Oberkochen, Germany) with 1000-, 2000- and 3000-times magnification. In addition, the surface free energy was evaluated according to Owens–Wendt–Rabel–Kaelble method by measuring the static and dynamic contact angles (OCA 15, DataPhysics Instruments GmbH, Filderstadt, Germany). To measure the coefficient of friction, a rounded diamond cone (cone angle 120°; 100 µm) was guided vertically over the titanium surfaces for a distance of 5 mm each.

### 2.3. Cleaning and Sterilization of the Titanium Discs

The test samples were initially cleaned in alcohol (70% ethanol) using an ultrasonic bath, followed by rinsing with distilled water (dH_2_O). Afterwards, samples were dried under laminar flow at room temperature for 30 min. Then, samples were prepared for sterilization. Without touching the conditioned and cleaned surface, the discs were steam sterilized in an autoclave at 134 °C for 5 min using the fractionated vacuum method.

### 2.4. Cell Cultivation

Immortalized human fetal osteoblasts (hFOB 1.19 cells; purchased from ATCC: LGC Standards GmbH, Wesel, Germany) were used for cell culture experiments. Cells were grown in a 1:1 mixture of Ham’s F12 and DMEM (without phenol red; Thermo Fisher Scientific, Dreieich, Germany) containing 10% FCS, 1% antibiotic-antimycotic solution and 0.3 mg/mL G418 selective antibiotics (standard medium; Thermo Fisher Scientific) at 34 °C in a humidified atmosphere and 5% CO_2_. Medium was changed every 2–3 days. Cells were routinely monitored for contamination with mycoplasma via PCR analysis and DAPI staining. To evaluate the influences of the different titanium surfaces on the cell proliferation, adhesion and osteoblastic differentiation, the sterilized titanium discs were placed in 24-well cell culture plates. Note that, 25.000–60.000 hFOBs per disc (long-term experiments) or 150.000 hFOBs per disc (2 h and 24 h experiments) were seeded in 200 µL culture medium. The cell suspension droplet was left on top of the titanium disc for 2 h to evaluate the initial adhesion. Then, the drop was aspirated, and the titanium discs with attached bone cells were either processed for further analysis (2 h time point) or were placed in new wells and cultured in 500 µL culture medium for longer time periods (24, 48, 72 and 7 d).

### 2.5. Crystal Violet Assay (CVA)

To quantify initial adhesion and proliferation of the hFOB cells on the titanium surfaces, crystal violet assays were performed, which are based on staining proteins and DNA of the attached cells (n = 5 for each time point). After the indicated time points, cultured cells on titanium discs were washed with phosphate-buffered saline (PBS), fixed by adding 4% paraformaldehyde (PFA) and incubated for 30 min at room temperature. After fixation, cells were washed with dH_2_O and stained for 1 h in an aqueous 0.05% crystal violet solution, washed twice with dH_2_O and air-dried. Finally, methanol was added to elute the dye from the stained cells and optical densities of eluates were measured at 540 nm.

### 2.6. MTT Assay

As a supporting approach to quantify adhesion and proliferation, the metabolic activity of osteoblasts was determined using the MTT assay (n = 5 for each time point). The assay is based on the reduction of water-soluble tetrazolium salt (MTT) by mitochondrial dehydrogenases into insoluble violet formazan dye. First, 40 µL of MTT stock solution (5 mg/mL in PBS) was added directly to medium in the wells with titanium discs and attached cells. After incubation at 37 °C for 4 h, 200 µL of cell culture medium was discarded and replaced by 200 µL of 10% (*w*/*v*) sodium lauryl sulfate in 0.01 M HCl. After gentle agitation, the plates were placed under laminar flow overnight at room temperature to elute the formazan dye from the cells. Next day, the medium of every well was transferred to 1.5 mL reaction tubes, and after centrifugation, optical density of the supernatants was determined at 570 nm.

### 2.7. Alkaline Phosphatase (ALP) Activity Assay

To evaluate the influence of titanium surface alterations on osteogenic maturation, the alkaline phosphatase (ALP) activity was measured after 7 and 12 d (n = 5 for each time point). Enzyme activities of hFOB 1.19 cell lysates were determined by measuring hydrolysis of p-nitrophenyl phosphate (p-NPP) to p-nitrophenyl (p-NP), as described elsewhere [30]. In brief, cultured cells on the titanium surfaces were washed with tris-buffered saline and cell lysis was conducted using ice-cold ALP lysis buffer (10 mM Tris-base, 1 mM MgCl_2_, 20 µM ZnSO_4_, 0.1% Triton X-100; pH 7.4) for 10 min on ice. Cell lysates were transferred to 1.5 mL reaction tubes following three freeze–thaw cycles and a centrifugation step. Finally, 100 µL of the supernatants was incubated with 50 µL p-NPP Substrate Buffer Solution (Sigma-Aldrich, Darmstadt, Germany) in 96-well microplates. After incubation at 37 °C, the level of p-NP production was measured by monitoring the optical density at 405 nm in a microplate reader. In addition, total protein concentrations in cell lysates were determined using the BCA protein assay kit (Thermo Fisher Scientific) according to the manufacturer’s protocol. Specific enzyme activities were calculated as amount of p-NP formed per minute with respect to the total protein concentration (U/mg protein).

### 2.8. Alizarin Red S Staining (ARS Assay)

Biomineralization of hFOB cells on titanium discs was quantified using the ARS assay (n = 5 for each time point). To induce osteogenic differentiation and biomineralization, bone cells seeded on titanium discs were cultured in osteogenic medium (standard medium with the addition of 50 mg/mL ascorbic acid and 10 mM β-glycerophosphate (Sigma Aldrich)) for 7 and 12 d. For the ARS assay, cells were washed twice with PBS and fixed in 4% PFA for 30 min at room temperature. After washing twice with dH_2_O, cells were incubated with 40 mM ARS solution (pH 4.1; Sigma-Aldrich, Munich, Germany) with gentle agitation for 30 min. Then, staining solution was removed and the mineralized matrices were washed 5× with dH_2_O. The remaining dye was eluted from the cell layer by adding 10% (*w*/*v*) cetylpyridine chloride in 10 mM sodium phosphate buffer (pH 7.0) to the wells with gentle agitation for 30 min. Subsequently, eluates were transferred to 1.5 mL reaction tubes, and after centrifugation, optical density of the supernatants was measured at 570 nm.

### 2.9. LDH Assay

The release of lactate dehydrogenase (LDH) into cell culture medium as a marker of cytotoxicity was measured using the Pierce LDH Cytotoxicity Assay kit (Thermo Fisher Scientific) (n = 5 for each time point). In brief, hFOBs were cultured on titanium discs for the indicated time points. Cells on titanium discs additionally treated with 1% Triton X-100 served as the positive control. Then, 50 µL cell culture medium per well was transferred to a new well of 96-well microplate. In addition, 50 μL of LDH reaction mixture was added to each well, and the plate was incubated for 30 min in the dark. The reaction was stopped by adding 50 μL of stop solution. Finally, the absorbance at 490 nm with correction wavelength at 680 nm was measured for every well, and cytotoxicity was calculated and normalized to positive controls, as recommended by the manufacturer.

### 2.10. Statistical Analysis

GraphPad Prism software, version 6, (GraphPad Software, San Diego, CA, USA), was used for statistical analysis. Mean ± standard error of the mean (SEM) were calculated, and one-way ANOVA and the post-hoc Tukey‘s multiple comparison were applied for CVA, MTT, ALP activity assay, Alizarin red staining and LDH assay. *p* values less than 0.05 were considered to be statistically significant.

## 3. Results

### 3.1. Surface Characterization

#### 3.1.1. Micro-Roughness

According to the SEM, all titanium surfaces exhibited a micro-rough topography characterized by irregular cavities of 1–3 µm in diameter that resulted from the acid etching procedure (Figure 1b,d,f,h,j). Topography analysis of the Ti6Al4V-surfaces examined via CLSM had the highest micro-roughness (Sz = 19.4 µm) in the standard-SA-surface group with F70 grain size and the standard blasting pressure (group 2.1), resulting in the most prominent enlargement of surface area (Sdr = 55.5 %) at the micron scale. Two-dimensional Sa values for mean arithmetic height in this group (Sa_F70 =_ 1.4 µm) and enlargement of surface area were within the commonly accepted ranges for moderately rough SA surfaces of 1–2 µm and 50%, respectively. Reduction in grain size (group 2.2 and 2.3) resulted in decreased values for Sa (Sa_F120_: 1.0 µm; Sa_F180_: 0.8 µm), Sz (Sz_F120_: 10.7 µm; Sz_F180_: 8.2 µm) and Sdr (Sdr_F120_: 34.4%; Sdr_F180_: 29.7%) when compared to the standard grain size (Table 2). In SEM pictures, all sandblasted surfaces clearly exhibited macro depressions whose diameters (default SA groups: ~20 µm) were closely correlated with blasting grain size (Figure 1), as is also observable in the parameter of peak density (Sds_F70_: 150 × 10^3^/mm^2^; Sds_F120_: 165 × 10^3^/mm^2^; Sds_F180_: 173 × 10^3^/mm^2^).

When compared to the standard SA surface, a reduction in blasting pressure or complete omission of blasting (i.e., etching only) resulted in lower Sa, Sz and Sdr values. Etched-only surfaces (group 2.5), as expected, had significantly lower Sa, Sz and Sdr values, classifying them as smooth–micro-rough surfaces according to Albrektsson and Wennerberg (Sa < 0.5 µm; Table 2). However, a reduction in blasting pressure by one third (group 2.4) only had minimal effects on topographical roughness parameters at the micron scale (Sa_2.4_: 1.2 µm; Sz_2.4_: 14.7 µm; Sdr_2.4_: 37.2%) (Table 2). In the SEM pictures, slightly flattened macro depressions are visible for this group; however, they visually closely resemble the standard SA group (2.1) in shape and diameter.

Concerning roughness parameters at the micron scale, all descriptive parameters were closely correlated to Sa values. If one group scored higher mean arithmetic height values, it would also score higher for all other recorded descriptive topographical parameters at the micron scale.

#### 3.1.2. Nano-Roughness

Using AFM, only slight differences in nano-roughness parameters were detected between the groups. However, a distinguishable nano-topography was not visible, and a linear connection between the blasting procedure and nano-topography was not apparent (data not shown). Sa values for mean arithmetic heights were smaller than 20 nm in all samples (Table 2).

#### 3.1.3. Physico-Chemical Properties

To assess the wettability of the titanium surfaces, static contact angles were measured using the sessile drop method and a contact angle instrument. All groups were considered hydrophobic (θ > 90°), and no significant differences in contact angles were detected between the groups. However, wettability was slightly higher for titanium surfaces blasted with default sized particles (2.1 and 2.4). Analysis of surface free energy between the treatment groups revealed differences in polar and disperse components of the surface energy, as a smaller polar component (γP) was detected in groups 2.2 (F120 grain size) and 2.4 (1/3 blasting pressure), when compared to the standard SA surface (γP_2.1_ = 4.3 mN/m; γP_2.2_ = 1.6 mN/m; γP_2.4_ = 1.3 mN/m).

Measurement of friction coefficient via scratch test yielded no significant differences.

### 3.2. Osteoblast Attachment and Viability

#### 3.2.1. Crystal Violet Assay

To determine initial adhesion onto the surfaces, cell proteins and cell DNA on the titanium discs were stained by crystal violet. After 2 h, osteoblasts on the Ti6Al4V discs that were treated with reduced blasting pressure (group 2.4) exhibited the lowest attachment rate when compared to the standard SA surface (Figure 2a).

After 72 h, etched-only discs exhibited the highest total cell numbers. Compared to all other groups, treatment with reduced blasting pressure (group 2.4) resulted in the lowest cell number at that time point (Figure 2g). The trend for lower cell numbers in discs treated with default size particles also appeared at the 7d time point, when significantly lower cell numbers were detected in those groups (2.1; 2.4) compared to the other treatment groups (Figure 2i). Cell proliferation was inversely correlated to grain size, and etched-only surfaces again exhibited the highest cell numbers of all examined treatments.

#### 3.2.2. MTT Assay

Complementary analysis on the adhesion and cell viability of the osteoblasts was performed by using MTT assays. Again, the etched-only group showed the highest metabolic activity at almost every time point (Figure 2). From the 48 h time point onwards, a significant trend for lower metabolic activity in the group subjected to decreased blasting pressure (group 2.4) was detected, when compared to all other groups (Figure 2).

#### 3.2.3. LDH Assay

LDH cytotoxicity assays was performed in cell culture supernatants, after 48 h, 72 h, 7 d and 12 d. In general, low cytotoxicity was detected in the groups, no matter the length of time (data not shown), with toxicity levels not exceeding 2.5%.

### 3.3. Osteoblast Differentiation

#### 3.3.1. Alizarin Red S Staining

To evaluate hFOB osteoblastic differentiation, biomineralization was investigated during maturation of cells. More mineralized extracellular matrix was detected on all titanium surfaces with time. No significant differences between the groups were detected within the cultivation period of 7 d, whereas after 12 d, significantly higher calcium deposition was detected on the etched-only Ti6Al4V surfaces (group 2.5) compared to all other surface groups (Figure 3). The surface sandblasted with small grains (F180) also exhibited higher biomineralization when compared to the other groups, except the etched-only group.

#### 3.3.2. ALP Activity Assay

Additionally, differentiation of hFOBs was investigated by observing alkaline phosphatase (ALP) activity after 7 and 12 d of cell culture on the titanium discs. ALP activity increased slightly during the cultivation period, and no significant differences were detected between the groups (data not shown).

### 3.4. Comparison of cpTi and Ti6Al4V

One aim of this study was to evaluate possible differences in osteoblast cell behavior on cpTi and Ti6Al4V SA surfaces, regarding biocompatibility. A comparison of default sandblasted cpTi (group 1) and Ti6Al4V (group 2.1) surfaces did not reveal significant differences regarding any of the above mentioned cytotoxic, adhesion, proliferation and differentiation parameters of hFOBs (Figure 4).

## 4. Discussion

So far, various studies have examined the effects of sandblasting procedures of implant surfaces on osseointegration [18,31,32,33,34]. However, even after decades of research, the influences of implant surface roughness and other surface parameters on the biological response remain a complex and controversial topic. Until now, it has just been vaguely defined what surface parameters reliably and definitely predict osteoblast cell behavior on titanium and titanium alloy implant surfaces [27,35].

Sandblasting with large-grit sand and acid-etching (SA) is one of the most widely applied surface treatments in clinical dental implantology. The procedure generates irregular depressions at the macroscale [36], up to 40 µm in diameter, through the sandblasting procedure, and the micron-scale topography is characterized by small micropits, 0.5–3 µm in diameter [37,38]. The surface has repeatedly demonstrated fast initial healing- and osseointegrative capabilities in vivo, when compared to other common implant surfaces [39,40,41].

The direct effect of sandblasting on osteoblast behavior is, however, hard to unravel in SA surfaces from in vivo studies, as the impact of the surface treatment on osseointegration could result from direct biological effects on osteoblasts, effects on other cell populations [34], enhanced mechanical fixation to bone [41,42,43,44] or other yet unknown secondary effects, including distance and contact osteogenesis [33]. On the other hand, the use of various different substrate materials, blasting molecules and etching procedures when generating SA surfaces also complicates the comparability and generalizability of relevant existing in vitro studies. Therefore, in this study the influences of a large-grit sandblasting procedure on osteoblast attachment, proliferation and differentiation on titanium alloy SA surfaces were evaluated in vitro by modification of blasting grain size and blasting pressure for implant surfaces created by an industrial procedure. Furthermore, osteoblast behavior was compared between cpTi SA and Ti6Al4V SA surfaces.

No differences regarding surface parameters or osteoblast behavior were found between cpTi and Ti6Al4V SA surfaces (Figure 4). In the Ti6Al4V alloy, the reduction in blasting grit size of corundum particles from 180–250 µm (F70) to 70 µm (F180) positively correlated with surface height parameters such as Ra, Sa, Sz and surface enlargement (Sdr) on the microscale. A reduction in blasting pressure by 1/3, however, only resulted in a small Sa change from 1.4 to 1.2 nm. Nano-topography (Sa: 9–12 nm) was considered smooth in all samples, and there were no relevant differences in surface free energy (18.5 ± 0.7 to 22.8 ± 0.7 mN*m^−1^) between the samples. Spatial parameters at the microscale such as summit density (Sds) inversely correlated with blasting grit size (Sds_2.1_ 150–Sds_2.3_ 173 mm^−2^). On SEM pictures (Figure 1), a reduction in the diameter of macroscale depressions from ~30 µm to <10 µm was apparent with the reduction of grit size.

So far, osteoblast cell behavior on titanium surfaces has been predominantly associated with arithmetic mean surface height parameters, such as one-dimensional Ra and two-dimensional Sa values [35].

A plethora of studies proposed influences of surface micro-roughness on osteoblast cell behavior: attachment, proliferation, differentiation and migration capabilities [45].

The majority of studies found an inverse relationship of osteoblast proliferation with micro-roughness [40,46,47,48,49,50,51,52,53,54]. However, there are studies that found the opposite- or inconclusive results [23,55,56,57,58,59,60,61,62] regarding proliferation and attachment. For osteoblast differentiation, many studies found enhanced differentiation of osteoblasts with increasing micro-roughness. However, there are studies that propose an optimal window for surface micro-roughness (Ra: 1–3 µm; 1–2 µm; [47,48]) or even found an inverse relationship between surface roughness and differentiation [46,49,55,58,61,62,63]. The controversial results of these studies are often not comparable per se, as they are for surfaces of different substrate materials generated by different procedures (blasting materials, coatings, etching solutions, etc.), which can have a substantial effect on osteoblast behavior [64]. Furthermore, experimental surfaces are often compared to machined or turned controls, which represent very smooth surfaces, and results thus should be generalized only with caution [62,65,66].

In our study, the observed effects on osteoblast behavior were likely the results of micro- and macro-topography, as we controlled for physicochemical characteristics such as wettability and nano-topography, and we applied consistent framework conditions for all samples (e.g., substrate material, blasting particles and subsequent etching procedure).

None of the titanium alloy discs in our study exhibited noticeable cytotoxicity. This is in agreement with studies on the biocompatibility of titanium and titanium alloys [67,68]. One limitation to our study is, however, that we did not control for chemical surface composition. The LDH assay and the fact that etching, which constitutes the last step in the manufacturing procedure, was the same in all samples, led us to the conclusion that surface chemistry was in fact similar for the samples [56,65,66]. We cannot, however, exclude the presence of aluminum remnants of the blasting procedure that might have affected the biological reaction of osteoblast-like cells [69]. We observed significantly reduced proliferation of osteoblasts on surfaces blasted with standard 180–250 µm grit (standard SA surface (group 2.1)) and SA surfaces treated with reduced blasting pressure (group 2.4), when compared to smaller-grit-size group and etched-only surfaces by crystal violet assay. Independently, MTT assay confirmed a significant reduction in osteoblast proliferation on SA surfaces treated with 180–250 µm grit and low blasting pressure (group 2.4), when compared to all other surfaces (Figure 2).

These results here are well in line with the narrative of reduced osteoblast proliferation with increasing surface roughness, as the surfaces that promoted weaker long-term proliferation also exhibited the top micro-roughness of all samples (group 2.1 (group 2.4): Sa 1.4 [1.2] µm). However, there was no apparent linear connection between Sa and osteoblast behavior in our study, so that the obtained results can only be generalized with extreme caution.

In contrast to studies that show enhanced osteoblast differentiation with increasing surface roughness, osteoblast differentiation in our study was significantly higher on the etched-only surface (Sa 0.3) when compared to all other surface treatments (Sa 0.8–1.4).

It has to be stated, however, that the literature on the direct comparison between etched-only and sandblasted and etched surfaces is scarce. Many of the studies that found increased osteoblast differentiation with higher surface micro-roughness—seeming to contradict our results—investigated machined or polished titanium surfaces and compared those groups with various different surface treatments that produce different levels of micro-roughness [40,46,51,57,64]. Studies, on the other hand, that directly compared the influence of etched-only surfaces to that of blasted and etched surfaces on osteoblast behavior [56,58,62,70] found higher osteoblast differentiation (Alizarin, 14d, primary rat bone marrow cells [71]) on etched cpTi surfaces when compared to blasted and etched surfaces. These results are corroborated by Conserva et al. [62], who found higher differentiation (ALP, 14d, SaOs-2) on etched-only surfaces when compared to SLA surfaces. Regarding proliferation and attachment, the groups of Rosalez-Leal et al. [56] and Keller et al. [70] found higher attachment on SLA after 1 h (MC3T3-E1 [70]), but lower proliferation after 24 and 48 h on SLA when compared to etched-only surfaces (MG63 [56]). Except for the study of Keller et al., who evaluated the osteoblast attachment at a single time point (1 h), our findings corroborate the results of those studies.

In the general context, our results indicate that increasing surface roughness at the micron scale per se is not suited to explaining proliferative and differentiation effects on osteoblasts, even when surface roughness parameters are within the acceptable boundaries [71]. These results may appear contradictory to the generally accepted narrative that increased surface micro-roughness, at least up to a certain limit, results in controlled changes in osteoblast proliferation or differentiation. We assume that the micron and submicron surface structure generated by etching-only is sufficient for osteoblast adhesion, proliferation and differentiation [26], even if the corresponding Sa and Ra, and other common descriptive surface parameters, are not within the proposed optimal ranges of moderately rough surfaces. We thus speculate that the sandblasting procedure on Ti6Al4V surfaces adds no direct beneficial effect to osteoblast cell behavior [56], and might rather positively affect other factors to improve osseointegration in vivo, such as macrophage or platelet behavior [43,72], protein absorption and mechanical stability [17], or by the generation of beneficial macrostructures for healing [3]. That might explain why in a direct comparison, SA surfaces exhibit higher removal torque, a surrogate for osseointegration at early time points in vivo, when compared to acid etched-only surfaces [41].

## 5. Conclusions

The present in vitro study revealed no significant differences in biocompatibility or osteoblast behavior between cpTi SA and Ti6Al4V SA surfaces. Variations in sandblasting grain size and blasting pressure on Ti6Al4V SA surfaces resulted in changes of the macro- and micro-topography, whereas the sandblasting procedure exhibited no relevant effect on nano-topography and hydrophilicity. Interestingly, micro-roughness parameters such as Sa did not affect osteoblast behavior, and omission of sandblasting (i.e., etching-only) resulted in similar or even greater osteoblast adhesion, proliferation and differentiation in vitro, when compared to the standard SA surface. On the other hand, only small differences in blasting grain size or blasting pressure resulted in a marked change in osteoblast behavior that could not be readily explained by surface roughness parameters. This work might contribute to better understanding the often entangled contributions to successful osseointegration of dental implants.

## Figures and Tables

**Figure 1 jfb-13-00143-f001:**
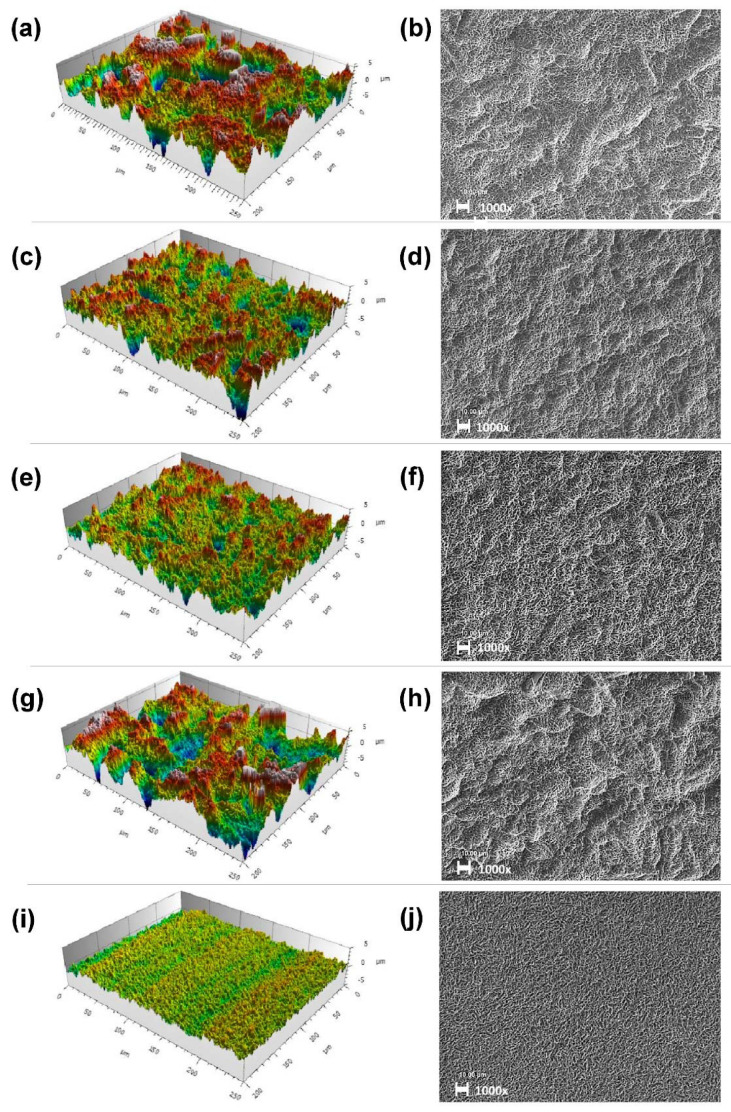
Surface topography via CLSM (measuring field: 250 µm × 200 µm; Gaussian filter 50 × 50 µm) (**a**,**c**,**e**,**g**,**i**) and respective SEM pictures (**b**,**d**,**f**,**h**,**j**) of Ti6Al4V surfaces. Group 2.1: (**a**,**b**); group 2.2: (**c**,**d**); group 2.3: (**e**,**f**); group 2.4: (**g**,**h**); group 2.5: (**i**,**j**). SEM pictures were taken at 1000 times magnification. Scale bars represent 10 µm.

**Figure 2 jfb-13-00143-f002:**
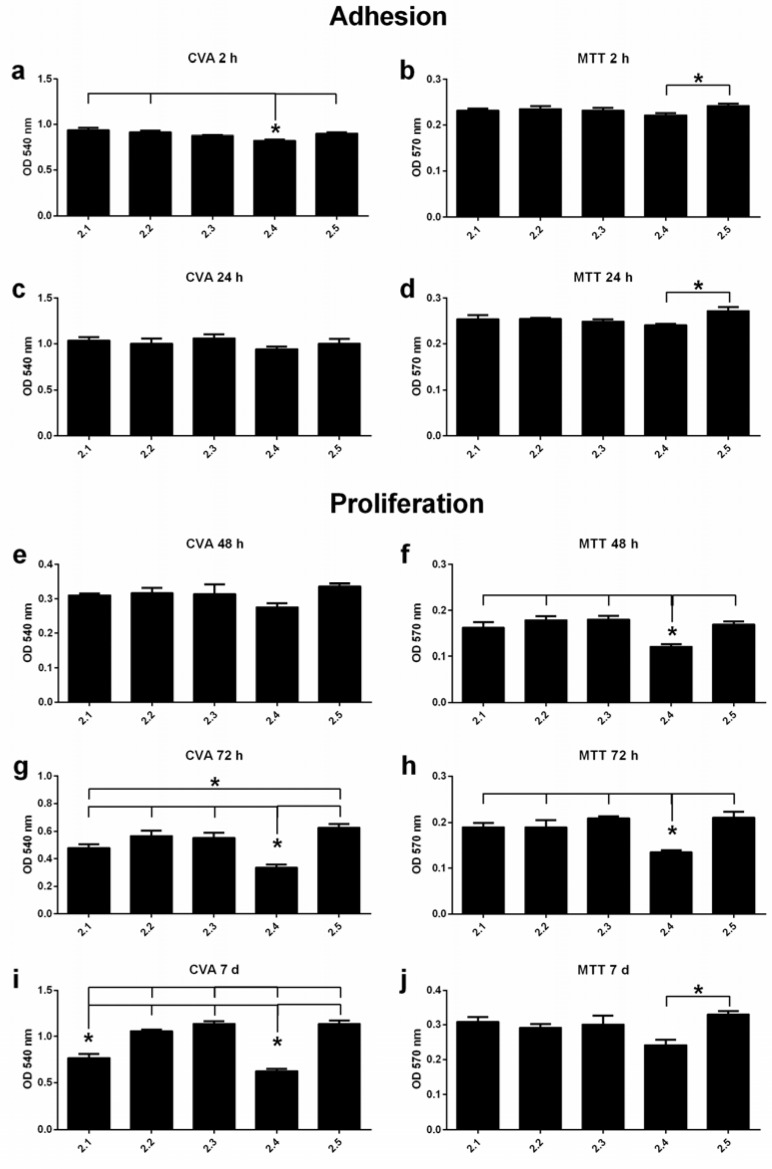
Adhesion and proliferation of hFOBs grown on Ti6Al4V surfaces at different time points (n = 5 for each time point). Crystal violet assays (CVA) (**a**,**c**,**e**,**g**,**i**) and MTT assays (**b**,**d**,**f**,**h**,**j**). Significant differences between the test groups are characterized by asterisks (*). OD-values are depicted as mean ± SEM.

**Figure 3 jfb-13-00143-f003:**
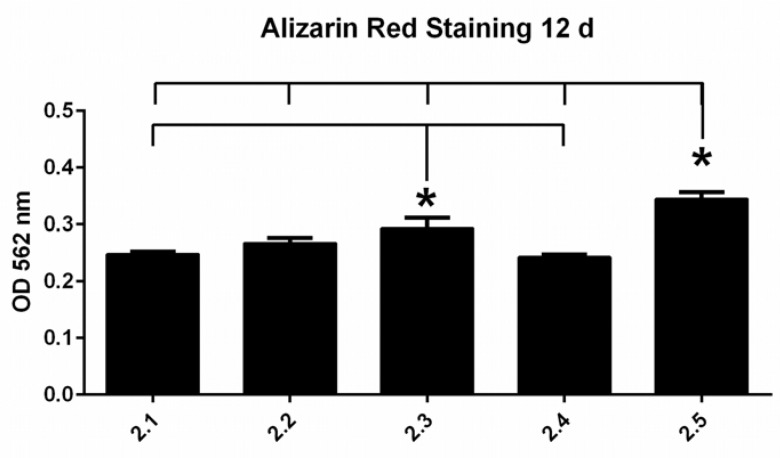
Biomineralization as evaluated by Alizarin Red staining after 12 days (n = 5). OD values represent arbitrary units. Significant differences between the groups are indicated with asterisks (*). OD values are depicted as mean ± SEM.

**Figure 4 jfb-13-00143-f004:**
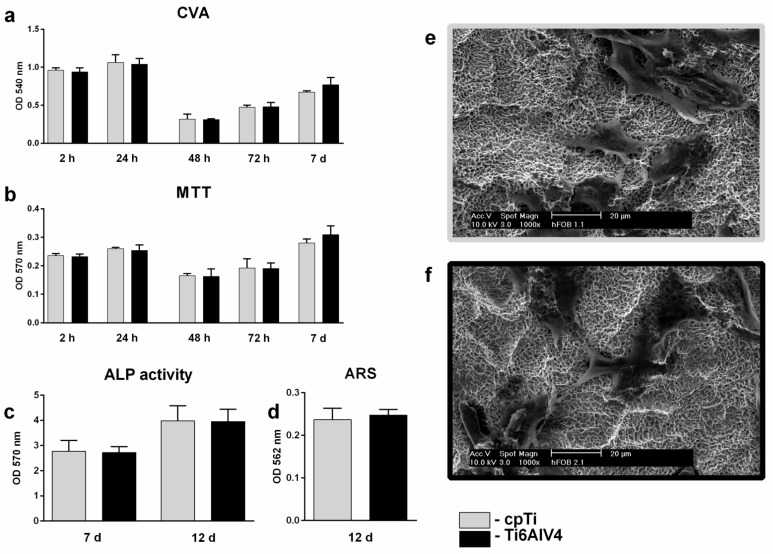
Comparison between cpTi- (group 1) and Ti6Al4V SA (group 2.1) surfaces regarding adhesion, proliferation and differentiation at different time points (n = 5 for each time point): crystal violet assay (**a**); MTT assay (**b**); ALP assay (**c**); Alizarin red staining (**d**). OD values are depicted as mean ± SEM. SEM pictures of hFOBs grown on titanium surfaces for 24 h are depicted in (**e**) (cpTi) and (**f**) (Ti6Al4V) at 1000× magnification. Scale bars represent 20 µm.

**Table 1 jfb-13-00143-t001:** Testing groups and respective surface modifications. Al_2_O_3_-particles were used as blasting material. All groups were subjected to acid etching after the blasting procedure. cpTi: commercially pure titanium, Ti6Al4V: titanium–aluminum–vanadium alloy, Std.: standardized pressure used for industrial manufacturing process by Dentaurum.

Group No.	Abbr.	Material	Blasting Grain Size	Pressure
1	4-F70S	cpTi	F70 [180–250 µm]	Std.
2.1	5-F70S	Ti6Al4V	F70 [180–250 µm]	Std.
2.2	5-F120S	Ti6Al4V	F120 [90–125 µm]	Std.
2.3	5-F180S	Ti6Al4V	F180 [63–90 µm]	Std.
2.4	5-F70R	Ti6Al4V	F70 [180–250 µm]	2/3 × Std.
2.5	5---	Ti6Al4V	-	-

**Table 2 jfb-13-00143-t002:** Roughness parameters of Ti6Al4V surfaces at the micro- and nano-scale and surface free energy. Sa: arithmetical mean height (microscale), Sz: mean roughness depth, Sq: rooth mean square height, Sdr: developed interfacial area ratio, Sds: density of summits, ϒ: surface free energy, Sa_n_: arithmetical mean height (nanoscale). The values are depicted as mean ± SD (n = 2).

Abbr.	Group No.	Sa [µm]	Sz [µm]	Sq [µm]	Sdr [%]	Sds [10^3^*mm^−2^]	Sa_n_ [nm]	ϒ [mN×m^−1^]
5-F70S	2.1	1.4 ± 0.04	17.9 ± 1.4	1.8 ± 0.08	55.5 ±2.8	150 ± 0.07	11 ± 0.3	21.85 ± 0.7
5-F120S	2.2	1.0 ± 0.05	10.7 ± 0.9	1.3 ± 0.06	34.4 ± 1.4	165 ± 0.14	12 ± 2.1	18.45 ± 0.7
5-F180S	2.3	0.8 ± 0.02	8.2 ± 0.2	1.1 ± 0.02	29.7 ± 0.6	173 ± 0.99	9 ± 2.2	22.4 ± 0.7
5-F70R	2.4	1.2 ± 0.01	14.7 ± 0.5	1.5 ± 0.04	37.2 ± 1.7	150 ± 0.35	11 ± 3.7	22.8 ± 0.7
5---	2.5	0.3 ± 0.01	3.4 ± 0.6	0.4 ± 0.01	18.4 ± 0.5	183 ± 0.92	12 ± 3.9	18.95 ± 0.7

## Data Availability

Not applicable.

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
