# Peer review of "Effects of Different Titanium Surface Treatments on Adhesion, Proliferation and Differentiation of Bone Cells: An In Vitro Study"

_jfb, 2022, doi:10.3390/jfb13030143_

Round 1

Reviewer 1 Report

The study reports the effect of various SLA and acid-etching treatments on the adhesion, proliferation, and biomineralization of human fetal osteoblasts. The results are interesting; however, before we move the manuscript to publication, the authors should address several issues.

1. The ‘Ra’ should be defined before use (Page 2 line 82).

2. The composition of the working gas and the exact gas pressure (‘Std.’) should be indicated in the M&M section (Page 3).

3. The morphology and size distribution (not just the particle size range) of the Al2O3-particles should be examined.

4. The procedures to determine the ‘Sdr’, ‘Sds’, and ’ϒ’ values should be detailed in the M&M section.

5. The composition of the acid and associated parameters for the etching should be detailed in the M&M section.

6. The culture duration for Figures 4e and 4f should be indicated in the figure caption. 

7. The initial adhesion pattern (first few hours after seeding) of the cells should be examined.

8. There are inconsistent descriptions in the text. For example, the blasting grain size for the std. group was ‘180-250 μm’ in Table 1 (Page 3); however, this parameter was described as ‘250-500 μm’ in the discussion section (such as Page 11, Line 419). The whole manuscript should be double-checked.

9. The number of measurements should be indicated in the captions for Figures 2, 3, and 4.

10. In Table 2, the data for Sa, Sz, Sq, Sdr, Sds, and San was likely from one measurement. More measurements and acceptable statistical analysis should be carried out.

11. Why group No.1 was not included in the surface topography, adhesion, proliferation, and biomineralization assays.

12. Since the etched-only surface induced better cell functions than that of the SLA groups, the authors should explain why the etched-only group has such effects.  Are they had different surface functional groups? More experiments should be done to uncover this aspect.

13. The effect of surface roughness on cell functions was found different from that reported by the others. Therefore, the author should discuss their results more carefully with details, for example, the differences in surface roughness range (including the test methods), cell line, cell concentration, culture duration, etc.

Reviewer 2 Report

I have a few comments for the authors to improve this paper. This paper is very clear and well written on a topic of interest. This is a classic topic in dental implants and the authors comparison here is noteworthy.

This paper made me think of the following review from Morgan Alexander. This is related to water contact tangle and not roughness, but similar idea. 10.1116/1.4989843 

Methods: Why was growth at 34C and not 37C?

Figure 1: Scale bar on SEM is quite difficult to see.

The authors need to be clearer and more specific as to what statistical tests were performed.

Reviewer 3 Report

Effects of different titanium surface…

The authors perform laboratory work on cell cultures. The purpose of this work is to compare both the material (CPTi /Ti6Al4V) and the surface treatment on cell adhesion, proliferation and differentiation. This study has a III-D level of evidence

As a result, the standard treatment has a better surface roughness (moderate roughness). This treatment presents 55.5% more surface area and a Sa of 1.4 while the reduction of the particulates means a reduction of the surface area.

Neither parameter affects wettability or coefficient of friction.

Both the pressure and the absence of sandblasting improve proliferation and differentiation.

Question: Page 10 lines 373-375 although the authors state that the effect of osseointegration is difficult to determine in in vivo studies due to the great diversity of factors involved, it is this interaction that finally determines the result, and not the individual component of each one of them. What is the authors' opinion?

I consider that this study is well designed, well executed and follows the scientific method. It is suitable for publication.

Round 2

Reviewer 1 Report

The manuscript has been improved and it is now good for publication.

Author Response

The manuscript has been improved and it is now good for publication.
We would again like to thank the reviewer for his comments and remarks, which have definitely improved our manuscript.